# CAT: Closed-loop Adversarial Training for Safe End-to-End Driving

**Linrui Zhang**[†], **Zhenghao Peng**[‡], **Quanyi Li**[§], **Bolei Zhou**[‡]
[†] Tsinghua University, [‡] UCLA, [§] The University of Edinburgh

**Abstract:** Driving safety is a top priority for autonomous vehicles. Orthogonal to prior work handling accident-prone traffic events by algorithm designs at the policy level, we investigate a **C**losed-loop **A**dversarial **T**raining (CAT) framework for safe end-to-end driving in this paper through the lens of environment augmentation. CAT aims to continuously improve the safety of driving agents by training the agent on safety-critical scenarios that are dynamically generated over time. A novel resampling technique is developed to turn log-replay real-world driving scenarios into safety-critical ones via probabilistic factorization, where the adversarial traffic generation is modeled as the multiplication of standard motion prediction sub-problems. Consequently, CAT can launch more efficient physical attacks compared to existing safety-critical scenario generation methods and yields a significantly less computational cost in the iterative learning pipeline. We incorporate CAT into the MetaDrive simulator and validate our approach on hundreds of driving scenarios imported from real-world driving datasets. Experimental results demonstrate that CAT can effectively generate adversarial scenarios countering the agent being trained. After training, the agent can achieve superior driving safety in both log-replay and safety-critical traffic scenarios on the held-out test set. Code and data are available at https://metadriverse.github.io/cat.

## 1 Introduction

While end-to-end driving has achieved promising performance in urban piloting [1] and track racing [2], safely handling accident-prone traffic events is one of the crucial capabilities to achieve for autonomous driving (AD). Benchmarking the safety and performance of an AI driving agent in simulation is a stepping stone for the real-world deployment [3]. However, it is insufficient to train or evaluate an end-to-end driving agents on traffic scenarios only retrieved from real-world traffic datasets [4, 5] since accident-prone events are extremely rare and difficult to collect in practice [6, 7].

Prior work improves the driving agent against safety-critical scenarios through various methods such as rule-based reasoning [8], motion verification [9], and constrained reinforcement learning [10]. Orthogonal to the elaborate algorithm designs at the policy level, recent studies obtain robust driving policies at the environment level by creating a set of accident-prone scenarios before hand as augmented training samples [11, 12]. Nevertheless, the learned policy may still easily overfit the fixed set of training samples thus fail to handle unknown hazards [13].

An alternate approach is to dynamically generate challenging scenarios that match the current capability of the driving agent being trained in a closed-loop manner. However, the state-of-the-art safety-critical scenario generation methods [11, 12, 14] are not yet applicable for that purpose due to the following issues: (i) *Scene generalizability*: probabilistic graph methods like CausalAF [11] require human prior knowledge of each scene graph and thus cannot scale to large and complex driving datasets; (ii) *Model dependency*: kinematics gradient methods like KING [12] relies on the forward simulation of the running policy and the backward propagation based on the environmental transition, which might not be accessible in the model-free end-to-end driving; (iii) *Time efficiency:*

7th Conference on Robot Learning (CoRL 2023), Atlanta, USA.

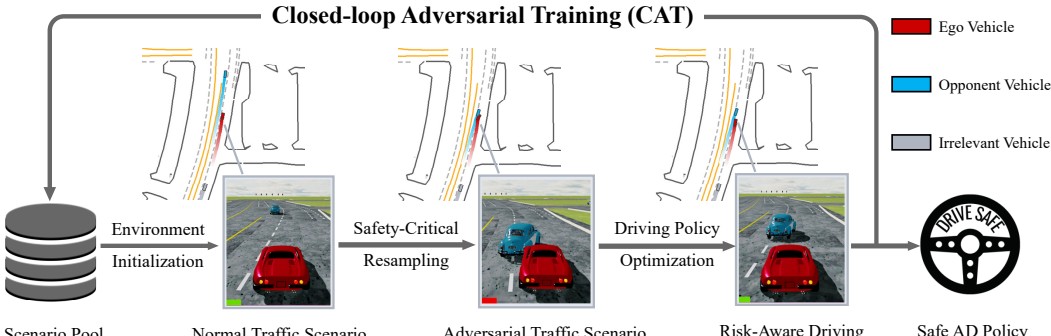

Figure 1: CAT iterates over safety-critical scenario generation and driving policy optimization in a closed-loop manner. In this example, the safety-critical resampling technique alters the behavior of the opponent vehicle (blue car) such that it suddenly cuts into the lane of the ego vehicle (red car), enforcing the agent to learn risk-aware driving skills such as deceleration and yielding.

autoregression-based generation methods like STRIVE [14] take minutes to optimize the adversarial traffic per scenario, which is time prohibitive for large-scale training with millions of episodes.

In this paper, we present the Closed-loop Adversarial Training (CAT) framework for safe end-to-end driving. As shown in Fig. 1, CAT imports driving scenarios from real-world driving logs and then generates safety-critical counterparts as adversarial training environments tailored to the current driving policy. The agent continuously learns to address emerging challenges and improves risk awareness in a closed-loop pipeline. CAT directly launches physical attacks against the estimated ego trajectory, the proposed framework is thus agnostic to the driving policy used by the agent and is compatible with a wide range of end-to-end learning approaches, such as reinforcement learning (RL) [15], imitation learning (IL) [16], and human-in-the-loop feedback (HF) [17].

One crucial component of the proposed framework is a novel factorized safety-critical resampling technique that efficiently turns logged driving scenarios into safety-critical ones during training. Specifically, we cast the safety-critical traffic generation as the risk-conditioned Bayesian probability maximization and then decompose it into the multiplication of standard motion forecasting sub-problems. Thus, we can utilize off-the-shelf motion forecasting models [18, 19] as the learned prior to generate adversarial scenarios with high fidelity, diversity, and efficiency. Compared to previous safety-critical traffic generation methods, the proposed technique obtains a competitive attack success rate while significantly reducing the computational cost, making the CAT framework effective and efficient for closed-loop end-to-end driving policy training.

To demonstrate the efficacy of our approach, we incorporate the proposed CAT framework into the MetaDrive simulator [20] and compose adversarial traffic environments from five hundred complex driving scenarios in a closed-loop manner to train RL-based driving agents without any ad-hoc safety designs. Experimental results show that CAT generates realistic and challenging physical attacks, and the resulting agent obtains superior driving safety in both log-replay and adversarial traffic scenarios on the held-out test set. The contributions of this paper are summarized as follows:

i) We propose an efficient safety-critical scenario generation technique by resampling the learned traffic prior, which improves attack success rate and lowers computation cost compared to prior work, making continuous adversarial scenario generation viable in closed-loop AD training.

ii) We present a closed-loop adversarial training framework for end-to-end safe driving based on the above technique and demonstrate the proposed framework substantially improves AI driving safety in complex testing scenarios imported from the real world.

## 2    Related Work

**Adversarial Training for Autonomous Driving.** Deep neural networks (DNNs), pervasively used in learning-based AD systems, are found vulnerable to adversarial attacks [21, 22]. Recent stud-

ies tend to manipulate the physical environment to generate realistic yet adversarial observation sequences from LiDAR inputs [23], camera inputs [24], and other physical-world-resilient objectives [25]. Compared to the above work focusing on perception, adversarial training for AD decision-making is much less explored. Ma et al. [26] first investigate the adversarial RL on a single autonomous driving scenario. Wachi [27] employs the multi-agent DDPG algorithm [28] to enforce the competition between player and non-player vehicles. In addition to algorithmic level designs, a more natural but less explored approach is to iteratively propose challenging scenarios during training [29]. There is a line of works on evolving training environments in RL [30, 31]. However, existing approaches are evaluated only in simplified environments like bipedal walker and heuristically modify the terrain or static barriers, which is not sufficient for complex AD tasks. In this work, we focus on generating realistic and safety-critical traffic scenarios to facilitate closed-loop adversarial training for end-to-end driving.

**Safety-critical Traffic Scenario Generation.** Safety-critical traffic scenario generation is of great value in adaptive stress testing [32] and corner case analysis [33] for the research and development of autonomous vehicles. L2C [34] learns to place and trigger a cyclist to collide with the target vehicle via RL algorithms, but it goes far to model complex vehicle interactions in real-world scenes. For robust imitation learning, kinematics gradients [12] and black-box optimization [23] can be used to magnify traffic risks. However, it relies on the forward simulation of the running policy and the backward propagation based on the vehicle kinematics, which might not be accessible in model-free end-to-end driving. CausalAF [11] builds scenario causal graphs to uncover behavior of interest and generates additional training samples to improve the robustness of driving policies. Nevertheless, the evaluations are limited to three scenarios since it requires human prior knowledge of each scene and thus hardly scale to a larger dataset. STRIVE [14] constructs a latent space to constrain the traffic prior and searches for the best responsive mapping via gradient-based optimization on that dense representation. Despite its impressive results on realistic traffic flows, the autoregression on raster maps takes several minutes to optimize the adversarial traffic for each scene, which brings about a costly computational burden for periodic policy optimization. We refer to the survey [35] for more detailed safety-critical scenario generation methodologies. Different from the above literature, we propose a novel adversarial traffic generation algorithm for real-world scenarios with an admissible time consumption, making it viable for large-scale policy iterations involving millions of episodes.

## 3 Method

In this section, we first formulate the closed-loop adversarial training (CAT) for safe end-to-end driving as a min-max problem in the context of RL, and then introduce the factorization of the learned traffic prior so as to generate adversarial driving scenarios efficiently in practice.

### 3.1 Problem Formulation

End-to-end driving directly uses raw sensor data as the inputs and outputs the low-level control command. Safe end-to-end driving incorporates risk-awareness into the above end-to-end pipeline and aims to minimize traffic accidents while maintaining the performance of route completion. We focus on reinforcement learning (RL)-based driving policy in this work, though the proposed CAT can be extended to accommodate a range of end-to-end driving policies. In our scope, the driving task can be formulated as Markov Decision Process (MDP) [36] in the form of $(S, A, R, f)$. $S$ and $A$ denote the state and action spaces, respectively. $S$ includes maps sensor readings such as camera images or LiDAR point cloud, high-level navigation commands and vehicle states. $A$ consists of low-level control commands like steering, throttle and brake. The reward function can be defined as $R = d - \eta c$, wherein $d$ is the displacement toward the destination, $c$ is a boolean value indicating collision with other objects and $\eta$ is a hyper-parameter for the reward shaping. $f$ is the transition function to describe the dynamics of the traffic scenario. The goal is to maximize the expected return $J(\pi, f) = \mathbb{E}_{\tau \sim \pi} \left[ \sum_{t=0}^{T} R(s_t, a_t) \right]$ the driving policy $\pi$ receives within the time horizon $T$, where $\tau \sim \pi$ is short handed for $a_t \sim \pi(\cdot|s_t), s_{t+1} \sim f(\cdot|s_t, a_t)$. CAT aims to enhance the robustness of

the learning agent via the following adversarial optimization:

$$\max_{\pi} \min_{f^{Adv} \in \mathcal{F}} J(\pi, f^{Adv}). \tag{1}$$

Here, the adversarial transition function $f^{Adv}$ must be within the feasible set $\mathcal{F}$ that is aligned with realistic traffic distribution, otherwise the learned driving policy $\pi$ is not applicable in practice.

The fundamental problem is to construct $f^{Adv}$ by generating compliant future traffic trajectories that are prone to collisions with the agent's rollouts. To formalize the traffic collision, we denote the vehicle controlled by the learning agent as the ego vehicle (Ego) and other vehicles as opponent vehicles (Op) and represent a traffic scenario as a tuple $(M, S_{1:T}^{\text{Ego}}, \boldsymbol{S}_{1:T}^{\text{Op}})$ with duration $T$ time steps. Here, the High-Definition (HD) road map $M$ consists of road shapes, traffic signs, traffic lights, etc. $S_{1:t}^{\text{Ego}}$ denotes the past states of the ego vehicle. $\boldsymbol{S}_{1:t}^{\text{Op}}$ is an $N$-element array $[S_{1:t}^{\text{Op}_1}, ..., S_{1:t}^{\text{Op}_N}]$, wherein each element stands for the past states of the corresponding opponent. For simplicity, we denote $X = (M, S_{1:t}^{\text{Ego}}, \boldsymbol{S}_{1:t}^{\text{Op}})$ as the information cutoff by step $t$ and $Y^{\text{Ego}} = S_{t:T}^{\text{Ego}}$, $\boldsymbol{Y}^{\text{Op}} = \boldsymbol{S}_{t:T}^{\text{Op}}$ are the future trajectories of ego and opponent starting from $t$, respectively. $Y^{\text{Ego}}$ is conditioned on the RL agent $\pi$. The cutoff step $t$ is fixed. We define a binary random variable $Coll = \{True, False\}$ to denote whether $Y^{\text{Ego}}$ collides with $\boldsymbol{Y}^{\text{Op}}$. Considering that the opponent vehicle must launch effective attacks based on the potential ego behavior which is responsive to the $\boldsymbol{Y}^{\text{Op}}$, the opponents' trajectories $\boldsymbol{Y}^{\text{Op}}$ and the ego vehicle's trajectory $Y^{\text{Ego}}$ are thus not independent. Therefore, we model $\boldsymbol{Y}^{\text{Op}}$ and $Y^{\text{Ego}}$ jointly and the safety-critical scenario distribution is expressed as:

$$\mathbb{P}(Y^{\text{Ego}}, \boldsymbol{Y}^{\text{Op}} | Coll = True, X) \tag{2}$$

Proposition 1 further shows that the construction of $f^{Adv}$ can be cast as marginal probability maximization of opponent trajectories $\boldsymbol{Y}^{\text{Op}}$ based on the above joint posterior distribution, where we assume that $Y^{\text{Ego}}$ generated by the current driving policy $\pi$ is sampled from $\mathcal{Y}(\pi)$.

**Proposition 1.** *Suppose that $\pi$ forces the agent to approach the destination and the episode terminates when any traffic collision happens, then we have*

$$\min_{f^{Adv} \in \mathcal{F}} J(\pi, f^{Adv}) \Leftrightarrow \max_{\boldsymbol{Y}^{Op}} \sum_{Y^{Ego} \sim \mathcal{Y}(\pi)} \mathbb{P}(Y^{Ego}, \boldsymbol{Y}^{Op} | Coll = True, X). \tag{3}$$

### 3.2 Factorized Safety-Critical Resampling

The joint distribution in Eq. (3) is still intractable. However, under the assumptions that the ego vehicle's reactions are unidirectionally based on the future traffic, we can factorize it with the Bayesian formula as shown in Proposition 2.

**Proposition 2.** *Suppose that $Y^{Ego}$ depends on $\boldsymbol{Y}^{Op}$ unidirectionally, then we have*

$$\mathbb{P}(Y^{Ego}, \boldsymbol{Y}^{Op} | Coll = True, X) \propto \mathbb{P}(\boldsymbol{Y}^{Op} | X) \mathbb{P}(Y^{Ego} | \boldsymbol{Y}^{Op}, X) \mathbb{P}(Coll = True | Y^{Ego}, \boldsymbol{Y}^{Op}). \tag{4}$$

After the factorization, we can search the best responsive $^*\boldsymbol{Y}^{\text{Op}}$ to magnify the probability of traffic collisions with the ego agent as possible through the marginal probability maximization given as:

$$
\begin{aligned}
&\max_{\boldsymbol{Y}^{\text{Op}}} \sum_{Y^{\text{Ego}} \sim \mathcal{Y}(\pi)} \mathbb{P}(Y^{\text{Ego}}, \boldsymbol{Y}^{\text{Op}} | Coll = True, X) \\
&= \max_{\boldsymbol{Y}^{\text{Op}}} \underbrace{\mathbb{P}(\boldsymbol{Y}^{\text{Op}} | X)}_{\text{1st Term}} \sum_{Y^{\text{Ego}} \sim \mathcal{Y}(\pi)} \underbrace{\mathbb{P}(Y^{\text{Ego}} | \boldsymbol{Y}^{\text{Op}}, X)}_{\text{2nd Term}} \underbrace{\mathbb{P}(Coll = True | Y^{\text{Ego}}, \boldsymbol{Y}^{\text{Op}})}_{\text{3rd Term}}.
\end{aligned}
\tag{5}
$$

It is beneficial to perform the above safety-critical traffic probability factorization since each term in Eq. (5) features a specific meaning and is tractable to handle. They are interpreted as follows:

i) **Traffic prior.** The 1st term is the standard motion prediction problem in which we can leverage arbitrary probabilistic traffic models [18, 37, 38, 39] to portray the multi-modal trajectory distribution. Taking the pre-trained model as the traffic prior enables the attack plausibility in complex scenarios without human specifications.

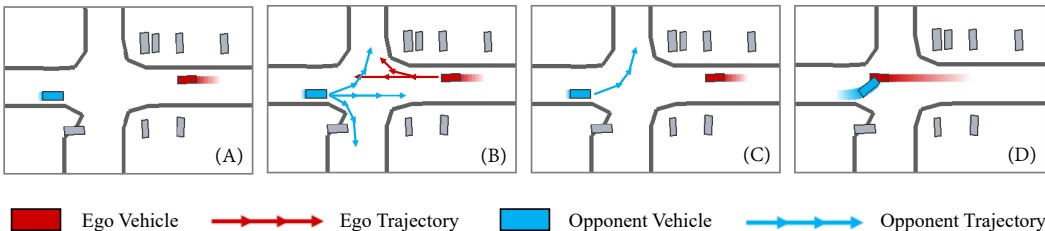

Figure 2: Illustration of Factorized Safety-Critical Resampling. (**A**) We initialize 1s traffic history with the dense map representation. (**B**) We then predict the traffic prior as well as the agent's reaction. (**C**) The most accident-prone trajectory of the opponent vehicle is selected. (**D**) The generated scene is thus expected to be safety-critical.

ii) **Ego estimation.** The 2nd term denotes the interactive ego trajectory yielding to the current state and upcoming traffic flow. The transition can be deterministic if the world model is learned or accessible under model-based settings [12]. As for the inference of real-world-compliant traffic flows, we can employ an interactive motion predictor [19] conditioned on known surrounding vehicles' trajectories to better reflects the ego compliance under risky interactions.

iii) **Collision likelihood.** The 3rd term reflects the likelihood of a collision in the compositional future, which can be simulated directly or treated as a binary classifier to fit [40].

As shown in Fig. 2, it is possible to approach the near-optimal adversarial trajectory via numerical optimization after each term is calculated.

### 3.3   Practical Implementation

We summarize the overall implementation of the CAT framework for safe end-to-end driving in **Algorithm 1**. Recalling the training objective of CAT in Eq. (1), we need to perform iterative optimization of policy learning and adversarial environment generation synchronously in a closed loop. The policy optimization can be achieved by arbitrary end-to-end driving policy learning approaches, e.g., a vanilla RL algorithm. Below, we focus on the adversarial environment generation, where we utilize the proposed factorized safety-critical resampling in Eq. (5). Note that we make a simplification in CAT by enforcing a single rival to launch the attack in each generated scene while simply maneuvering the other vehicles to avoid self-collisions. This is reasonable since most traffic accidents are caused by two traffic participants rather than involving multiple vehicles.

We first predict the traffic prior $\mathbb{P}(Y^{\text{Op}}|X)$ using a pre-trained probabilistic traffic forecasting model $\mathcal{G}$. Considering the strong performance and the ease of sampling, we adopt DenseTNT [18], an anchor-free goal-based motion predictor, in this work. Specifically, we propose $M$ possible candidates $\{(Y_i^{\text{Op}}, P_i^{\text{Op}})\}_{i=1}^M$ in parallel. The component $Y_{i,k}^{\text{Op}}$ in the $k$-th time step consists of the predicted position and yaw of the opponent vehicle. The probability of the trajectory $P_i^{\text{Op}}$ coincides with the probability of the corresponding destination goal.

We then tackle the ego estimation term $\mathbb{P}(Y^{\text{Ego}}|Y^{\text{Op}}, X)$. Considering the non-stationary policy during training, we notice that the ego behavior does not necessarily match the logged behavior in the dataset. Consequently, directly utilizing the pre-trained traffic estimator derived from natural traffic flows [19] to provide ego trajectory probability has a severe bias. Alternatively, we record the latest $N$ rollouts of the ego vehicle in each scenario formed as $\{(Y_j^{\text{Ego}}, P_j^{\text{Ego}})\}_{j=1}^N$ wherein we derive the likelihood of visited state sequences deduced by the current policy $\pi$: $P_{j,k+1}^{\text{Ego}} = P_{j,k}^{\text{Ego}} \cdot \pi(a_k|s_k)$.

At last, we empirically estimate the collision likelihood $\mathbb{P}(Coll|Y^{\text{Ego}}, Y^{\text{Op}})$. Given the specific compositional future of $Y_j^{\text{Ego}}$ and $Y_i^{\text{Op}}$, we compute the minimal distance between their bounding boxes in the following steps and set the collision likelihood as $P_{i,j}^{Coll} = \alpha^k$ if the closest gap is $\leq 0$ at timestep $k$. If the collisions happen at multiple step, the earliest $k$ will be used. Here, $\alpha \in (0, 1]$ is a heuristic decay factor to reflect the increasing uncertainty of the traffic model.

**Algorithm 1:** Closed-loop Adversarial Training (CAT) for Safe End-to-End Driving.

---

**Input:** Initial driving policy $\pi$, learning algorithm $\mathcal{T}$, trajectory predictor $\mathcal{G}$, the simulator.
**Output:** Robust driving policy $\pi^*$.

1 Initialize the scenario pool $\mathcal{D} = \{X_1, X_2, ...X_{|\mathcal{D}|}\}$ from real-world datasets.
2 Initialize the ego trajectory buffer for each scenario.
3 **while** $\pi$ *is not converged* **do**
4      Randomly sample a logged traffic $X$ from the scenario pool $\mathcal{D}$.
5      Retrieve the ego trajectory buffer for this scenario $\{(Y_i^{\text{Ego}}, P_i^{\text{Ego}})\}_{i=1}^N$.
6      $\{(Y_i^{\text{Op}}, P_i^{\text{Op}})\}_{i=1}^M \sim \mathcal{G}(X)$      ▷ Generate the traffic prior, $M$ Op's trajectories.
7      **for** *i in* $1, 2, ..., M$ **do**                         ▷ For each Op candidate.
8          **for** *j in* $1, 2, ..., N$ **do**                   ▷ For each Ego candidate.
9               $P_{ij}^{Coll} = \begin{cases} \alpha^k & \text{if BBox}(Y_{j,k}^{\text{Ego}}) \text{ collides with BBox}(Y_{i,k}^{\text{Op}}) \text{ at step } k, \\ 0 & \text{otherwise.} \end{cases}$
10          $P(Y_i^{\text{Op}}|\pi, Coll, X) = P_i^{\text{Op}} \sum_{j=1}^N P_j^{\text{Ego}} P_{ij}^{Coll}$      ▷ Compute the posterior probability.
11      $Y^{\text{Op}*} = \arg\max_{Y_i^{\text{Op}}} P(Y_i^{\text{Op}}|\pi, Coll, X)$         ▷ Select the best Op's trajectory.
12      obs = simulator.reset($X, Y^{\text{Op}*}$)      ▷ Reset sim to replay the adversarial scenario.
13      Initialize $Y^{\text{Ego}} = \{\}, P^{\text{Ego}} = 1.$ **for** *t in* $1, 2, 3..., |T|$ **do**    ▷ Rollout the policy against the adversarial scenario.
14          act $\sim \pi(\cdot|\text{obs})$
15          obs = simulator.step(act)
16          $Y^{\text{Ego}} \leftarrow Y^{\text{Ego}} \bigcup \{Y_t^{\text{Ego}}\}$                      ▷ Update Ego trajectory.
17          $P^{\text{Ego}} \leftarrow P^{\text{Ego}} \cdot \pi(\text{act}|\text{obs})$               ▷ Update Ego probability.
18      $\pi \leftarrow \mathcal{T}(\pi)$                                   ▷ Policy optimization.
19      Add $(Y^{\text{Ego}}, P^{\text{Ego}})$ to the ego trajectory buffer for this scenario.

---

## 4 Experiments

### 4.1 Experiment Setup

We import 500 real-world traffic scenarios involving complex vehicle interactions from the Waymo Open Motion Dataset (WOMD) [4] as the raw data. Each scene in WOMD contains a traffic participant labeled as *Object of Interest* regarding the ego car, which is also designated as the opponent vehicle in our experiments. All the experiments are conducted in MetaDrive [20], an open-source and lightweight AD simulator. The specific state, action and reward function in policy training and detailed hyper-parameter settings in safety-critical scenario generation are placed in Appendix C and D. Here, we point out some pivotal parameters. Each scene lasts 9s, in which we take the first 1s traffic history as $X$ and manipulate the following 8s to generate the adversarial trajectory $Y^{\text{Op}}$. We set $M = 32$ as the number of opponent trajectory candidates, $N = 5$ as the length of ego rollout queue and $\alpha = 0.99$ to penalize the uncertainty of motion forecasting.

### 4.2 Evaluation of Safety-critical Traffic Generation in CAT

The factorized safety-critical resampling is the crucial component of CAT to generate adversarial training samples. We provide qualitative and quantitative comparisons with the following baselines: **(A) Raw Data**: Replaying the recorded real-world traffic. **(B) M2I (adv)** [19]: The interactive traffic motion prediction is similar to our factorized formulation and thus can be modified as an adversarial scenario generator. **(C) STRIVE** [14]: The state-of-the-art safety-critical scenario generation methods performing gradient-based optimization on latent variables.

**Qualitative analysis.** In Fig. 3, we present 9 different types of safety-critical scenarios that CAT generates from raw scenes, according to the pre-crashed traffic categorized by the National Highway Traffic Safety Administration (NHTSA). It can be concluded that CAT is able to generate adversarial traffic given arbitrary real-world raw scenes. Meanwhile, the generated trajectories are in line with human driver behavior, even though we don't specify prior knowledge of that scene. In Fig. 4, we

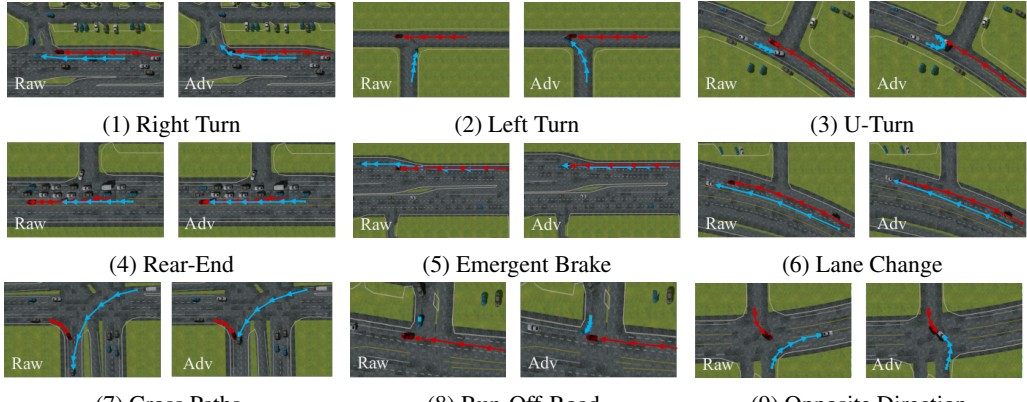

(1) Right Turn         (2) Left Turn         (3) U-Turn

(4) Rear-End      (5) Emergent Brake      (6) Lane Change

(7) Cross Paths      (8) Run-Off-Road    (9) Opposite Direction

Figure 3: Qualitative results on the diversity of safety-critical scenarios generated by CAT. In each subfigure, the left and right are the raw scene and the adversarial counterpart. The ego and adversarial trajectories are highlighted with red and blue arrows, respectively.

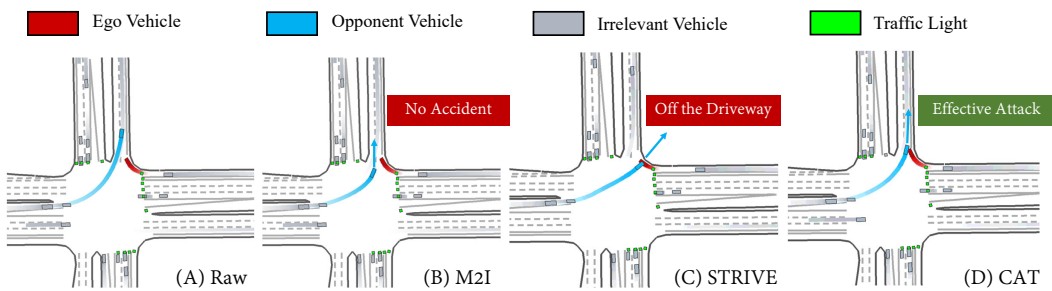

(A) Raw        (B) M2I        (C) STRIVE        (D) CAT

Figure 4: Qualitative results on the plausibility of safety-critical scenarios generated by CAT. The attack is regarded as effective only if leading traffic accidents are consistent with real-world events.

compare the generated adversarial traffic of the four methods on the same intersection. In the raw scene, the leading vehicle turns preferentially and does not cross the path of the ego vehicle. The opponent attempts to collide with the agent at the intersection through the safety-critical generation. However, M2I (adv) has a bias in estimating the reaction of the ego vehicle, which does not cause the expected accident. STRIVE finds the solution to enforce a crash, but it is still cumbersome to tweak the multinomial loss function to balance the goal of colliding as soon as possible and reasonable driving behavior, like keeping the vehicle in the driveway. By contrast, our factorized safety-critical resampling leverages the learned motion prior to regularize the opponent's trajectory, magnifying the traffic risk while preserving its plausibility. More visualization can be found in Appendix E.

**Quantitative analysis.** In Tab. 1, we compare adversarial traffic generation methods on 100 test scenes, focusing on two metrics. The first metric of interest is the attack success rate as the driving policies are responsive and even defensive to the traffic flow. We adopt three kinds of agents with fixed policies to validate: *(i) Replay Agent*: Replay the original trajectory of the ego ve-

Table 1: Comparing adversarial generation methods.

| Methods | Attack Success Rate ↑ | | | Per Scene Generation Time ↓ |
|---|---|---|---|---|
| | Replay | IDM | Pretrained | |
| Raw Data | 0% | 34% | 14% | / |
| M2I (adv) | 47% | 41% | 19% | **0.41 ± 0.03**s |
| STRIVE | 85% | 82% | 66% | 153.10 ± 47.33s |
| CAT ($N = 1$) | 91% | 71% | 62% | 0.66 ± 0.09s |
| CAT ($N = 5$) | **91**% | **86**% | **69**% | 3.34 ± 0.41s |

hicle logged in real-world data-set. *(ii) IDM Agent*: A heuristic controller well-adopted in AD tasks [41]. *(iii) Pre-trained Agent*: A pre-trained RL policy on WOMD. We find that M2I (adv) is insufficient for ego prediction and attacks less effectively especially against low-level policy, which is fatal for end-to-end driving. CAT collects ego rollouts to enhance the confidence of ego estimation during training ($N = 5$) and testing ($N = 1$) which significantly improves the attack success rate and is competitive with the SOTA method STRIVE. The second metric of interest is the time consumption per scene, which is non-negligible considering the large number of scenario iterations

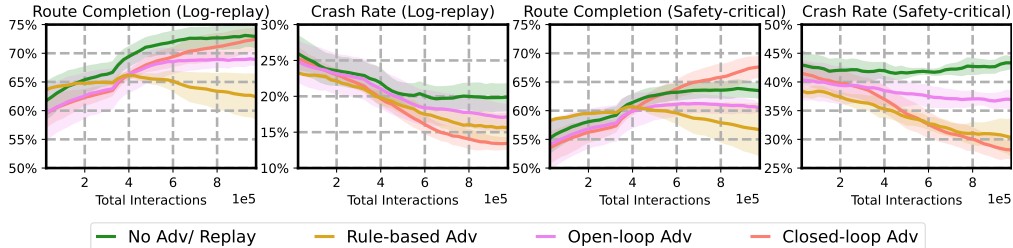

Figure 5: The learning curves of the policies trained with different pipelines.

Table 2: Performance of driving policies with different training pipelines on the held-out test set.

| Methods | Log-replay Scenarios | | Safety-critical Scenarios | |
|---------|---------------------|---|--------------------------|---|
| | Route Completion ↑ | Crash Rate ↓ | Route Completion ↑ | Crash Rate ↓ |
| No Adv/ Replay | **72.91% ± 2.05%** | 19.89% ± 1.95% | 63.48% ± 1.46% | 43.33% ± 1.13% |
| Rule-based Adv | 62.42% ± 3.99% | 15.61% ± 1.98% | 56.68% ± 4.66% | 30.31% ± 3.33% |
| Open-loop Adv | 68.89% ± 1.05% | 17.15% ± 1.80% | 63.48% ± 1.46% | 36.96% ± 1.66% |
| Closed-loop Adv | 72.47% ± 2.04% | **13.43%±0.88%** | **67.62% ± 1.89%** | **28.15%±1.63%** |

during training. We find that STRIVE generally requires 2-3 minutes to process a single scene due to its autoregression procedure on the raster map, which means it takes days to train the agent in a closed loop involving thousands of episodes. By contrast, our approach best balance the attack success rate and computational time compared and admits a privileged advantage in closed-loop adversarial training for end-to-end driving.

## 4.3 Evaluation of Closed-loop Adversarial Training in CAT

We show how the driving agent improves its safety performance within CAT framework. We split the 500 raw scenes into 400 training and 100 testing scenarios. We train a TD3 [42] driving policy from scratch with 4 types of training pipelines: **(A) No Adv/ Replay**: The raw driving scenarios are used as the training environments. **(B) Rule-based Adv**: We implement a rule-based system that overwrites the trajectories in data to generate physical attacks (see the Appendix F for details). **(C) Open-loop Adv**: We generate the opponent trajectories that collide with the ego trajectories against the log-replayed ego rollout before training. **(D) Closed-loop Adv**: We use CAT to generate adversarial scenario on-the-fly against the ego trajectories generated by the learning agent.

We evaluate the driving policies trained from different pipelines with two metrics. The first metric is the route completion rate, which measures the progress the agent makes; The second metric is the crash rate, the ratio of episodes that the ego vehicle crashes into others. We first evaluate the policy on the held-out testing scenarios with logged traffic (*Log-replay Scenarios*). Then we run CAT against the policy to generate adversarial traffic. Finally, we run the policy in the testing scenarios with CAT-generated traffic (*Safety-critical Scenarios*). As shown in Table 2 and Fig. 5, we find that CAT substantially enhances safety performance compared with vanilla RL training, reducing crash rate by 6.46% in log-replayed scenarios and 15.18% in safety-critical ones with competitive route completion. More qualitative results can be referred in Appendix G. Besides, we demonstrate that generating adversarial environments against current policy on-the-fly makes the trained policy performs better. At last, factorized safety-critical resampling can preserve the realistic traffic distribution so the learned policy has competitive route completion rate. On the contrary, the rule-based attacks lead to over-conservative driving policy that has inferior route completion.

## 5 Conclusion

In this paper, we investigate how to improve the safety of end-to-end driving through the lens of safety-critical traffic scenario augmentation. Empirical results demonstrate that the proposed closed-loop adversarial training (CAT) framework can provide realistic physical attacks efficiently during training and enhance AI driving safety performance in the test time.

**Acknowledgments**

This work was supported by the National Science Foundation under Grant No. 2235012 and the Cisco Faculty Award.

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

# A  Proof of Proposition 1

**Proposition.** *Suppose that $\pi$ forces the agent to approach the destination and the episode terminates when any traffic collision happens, then we have*

$$\min_{f^{Adv} \in \mathcal{F}} J(\pi, f^{Adv}) \Leftrightarrow \max_{\boldsymbol{Y}^{Op}} \sum_{Y^{Ego} \sim \mathcal{Y}(\pi)} \mathbb{P}(Y^{Ego}, \boldsymbol{Y}^{Op} | Coll = True, X). \tag{A.1}$$

*Proof.* According to the definition of return $J(\pi)$ and reward function $R = d - \sigma c$, we have

$$\min_{f^{Adv} \in \mathcal{F}} J(\pi, f^{Adv}) \Leftrightarrow \min_{f^{Adv} \in \mathcal{F}} \sum (d - \sigma c) \tag{A.2}$$

Since $\pi$ forces the agent to approach the destination and the episode terminates when any traffic collision happens, $J$ is minimized when encountering collisions; otherwise $J = \sum d$ reaches its upper bound. Considering that the construction of $f^{Adv}$ is to maneuver the surrounding vehicles when the map is given, it equals that we search the best constraint-satisfying $\boldsymbol{Y}^{Op}$ in the prior trajectory distribution. Thus, we have

$$\min_{f^{Adv} \in \mathcal{F}} J(\pi, f^{Adv}) \Leftrightarrow \max_{\boldsymbol{Y}^{Op}} \ \mathbb{P}(\boldsymbol{Y}^{Op} | X) \quad \text{s.t Ego } controlled\ by\ \pi\ collides\ with\ \text{Op}. \tag{A.3}$$

We then rewrite Eq. (A.3) in the form of posterior probability distribution maximization as

$$\min_{f^{Adv} \in \mathcal{F}} J(\pi, f^{Adv}) \Leftrightarrow \max_{\boldsymbol{Y}^{Op}} \ \mathbb{P}(\boldsymbol{Y}^{Op} | \pi, Coll = True, X) \tag{A.4}$$

Suppose that $Y^{Ego}$ generated by the current driving policy $\pi$ can be sampled from $\mathcal{Y}(\pi)$, Eq. (A.4) is equivalent to marginal maximization over the joint trajectory distribution, which follows as

$$\min_{f^{Adv} \in \mathcal{F}} J(\pi, f^{Adv}) \Leftrightarrow \max_{\boldsymbol{Y}^{Op}} \sum_{Y^{Ego} \sim \mathcal{Y}(\pi)} \mathbb{P}(Y^{Ego}, \boldsymbol{Y}^{Op} | Coll = True, X). \tag{A.5}$$

The proof of Proposition 1 is completed. $\qquad\square$

# B  Proof of Proposition 2

**Proposition.** *Suppose that $Y^{Ego}$ depends on $\boldsymbol{Y}^{Op}$ unidirectionally, then we have*

$$\mathbb{P}(Y^{Ego}, \boldsymbol{Y}^{Op} | Coll = True, X) \propto \mathbb{P}(\boldsymbol{Y}^{Op} | X) \mathbb{P}(Y^{Ego} | \boldsymbol{Y}^{Op}, X) \mathbb{P}(Coll = True | Y^{Ego}, \boldsymbol{Y}^{Op}). \tag{B.1}$$

*Proof.* According to Bayes theorem, we have

$$\mathbb{P}(Y^{Ego}, \boldsymbol{Y}^{Op} | Coll = True, X) \propto \mathbb{P}(Coll = True | Y^{Ego}, \boldsymbol{Y}^{Op}, X) \mathbb{P}(Y^{Ego}, \boldsymbol{Y}^{Op}, X) \tag{B.2}$$

Since $Coll$ merely depends on $Y^{Ego}$ and $\boldsymbol{Y}^{Op}$, (B.2) is equivalent to

$$\mathbb{P}(Y^{Ego}, \boldsymbol{Y}^{Op} | Coll = True, X) \propto \mathbb{P}(Coll = True | Y^{Ego}, \boldsymbol{Y}^{Op}) \mathbb{P}(Y^{Ego}, \boldsymbol{Y}^{Op}, X) \tag{B.3}$$

Since we assume that $Y^{Ego}$ depends on $\boldsymbol{Y}^{Op}$ unidirectionally; continuing with Bayes theorem, we have

$$\begin{aligned} \mathbb{P}(Y^{Ego}, &\boldsymbol{Y}^{Op} | Coll = True, X) \\ &\propto \ \mathbb{P}(Coll = True | Y^{Ego}, \boldsymbol{Y}^{Op}) \mathbb{P}(Y^{Ego} | \boldsymbol{Y}^{Op}, X) \mathbb{P}(\boldsymbol{Y}^{Op}, X) \\ &\propto \ \mathbb{P}(Coll = True | Y^{Ego}, \boldsymbol{Y}^{Op}) \mathbb{P}(Y^{Ego} | \boldsymbol{Y}^{Op}, X) \mathbb{P}(\boldsymbol{Y}^{Op} | X) \mathbb{P}(X) \end{aligned} \tag{B.4}$$

Since the past state $X$ is given, we can omit the last item $\mathbb{P}(X)$ in (B.4). Therefore, it holds that

$$\mathbb{P}(Y^{Ego}, \boldsymbol{Y}^{Op} | Coll = True, X) \propto \mathbb{P}(\boldsymbol{Y}^{Op} | X) \mathbb{P}(Y^{Ego} | \boldsymbol{Y}^{Op}, X) \mathbb{P}(Coll = True | Y^{Ego}, \boldsymbol{Y}^{Op}) \tag{B.5}$$

The proof of Proposition 2 is completed. $\qquad\square$

## C   RL Experimental Settings

We implement CAT in MetaDrive [20]. MetaDrive simulator provides off-the-self RL environments for end-to-end driving. We follow the basic setting in MetaDrive[1].

In MetaDrive RL environments, the state includes maps sensor readings (Camera or LiDAR), high-level navigation command and self vehicle states. In our experiments, we use 2D LiDAR as the sensor to detect the surrounding vehicles, road boundaries and road lines. The state vector consists of three parts:

- Ego State: current states such as the steering, heading, velocity. (ii) Navigation: the navigation information that guides the vehicle toward the destination. Concretely, MetaDrive first computes the route from the spawn point to the destination of the ego vehicle.

- Navigation: the navigation information that guides the vehicle toward the destination. Concretely, MetaDrive first computes the route from the spawn point to the destination of the ego vehicle. Then a set of checkpoints are scattered across the whole route with certain intervals. The relative distance and direction to the next checkpoint and the next next checkpoint will be given as the navigation information.

- Surrounding: the surrounding information is encoded by a vector containing the Lidar-like cloud points. We use 72 lasers to scan the neighboring area with radius 50 meters.

The action consists of low-level control commands like steering, throttle and brake. MetaDrive receives normalized action as input to control each target vehicle: $\mathbf{a} = [a_1, a_2]^T \in [-1, 1]^2$. At each environmental time step, MetaDrive converts the normalized action into the steering $u_s$ (degree), acceleration $u_a$ (hp) and brake signal $u_b$ (hp) in the following ways: (i) $u_s = S_{max}a_1$, (ii) $u_a = F_{max} \max(0, a_2)$ , (iii) $u_b = -B_{max} \min(0, a_2)$, wherein $S_{max}$ (degree) is the maximal steering angle, $F_{max}$ (hp) is the maximal engine force, and $B_{max}$ (hp) is the maximal brake force.

MetaDrive uses a compositional reward function as $R = R_{driving} + R_{crash\_vehicle\_penalty} + R_{out\_of\_road\_penalty}$. Here, the driving reward $R_{driving} = d_t - d_{t-1}$, wherein the $d_t$ and $d_{t-1}$ denote the longitudinal coordinates of the target vehicle in the current lane of two consecutive time steps, providing dense reward to encourage agent to move forward. By default, the penalty is -1 if the agent collides with surrounding vehicles, and the penalty is -10 if the agent runs out of the road.

## D   Hyper-parameter Settings

| Table 3: CAT | | Table 4: TD3 | | Table 5: DenseTNT and M2I | |
|---|---|---|---|---|---|
| Hyper-parameter | Value | Hyper-parameter | Value | Hyper-parameter | Value |
| Scenario Horizon $T$ | 9s | Discounted Factor $\gamma$ | 0.99 | Train Batch size | 256 |
| History Horizon $t$ | 1s | Train Batch Size | 256 | Train Epochs | 30 |
| # of OV candidates $M$ | 32 | Critic Learning Rate | 3E-4 | Sub Graph Depth | 3 |
| # of EV candidates $N$ | 5 | Actor Learning Rate | 3E-4 | Global Graph Depth | 1 |
| Penalty Factor $\alpha$ | 0.99 | Policy Delay | 2 | NMS Threshold | 7.2 |
| Policy Training Steps | 10E6 | Target Network $\tau$ | 0.005 | Number of Mode | 32 |

---

[1]https://metadrive-simulator.readthedocs.io/en/latest/index.html

# E  Qualitative Results of Safety-critical Traffic Generation

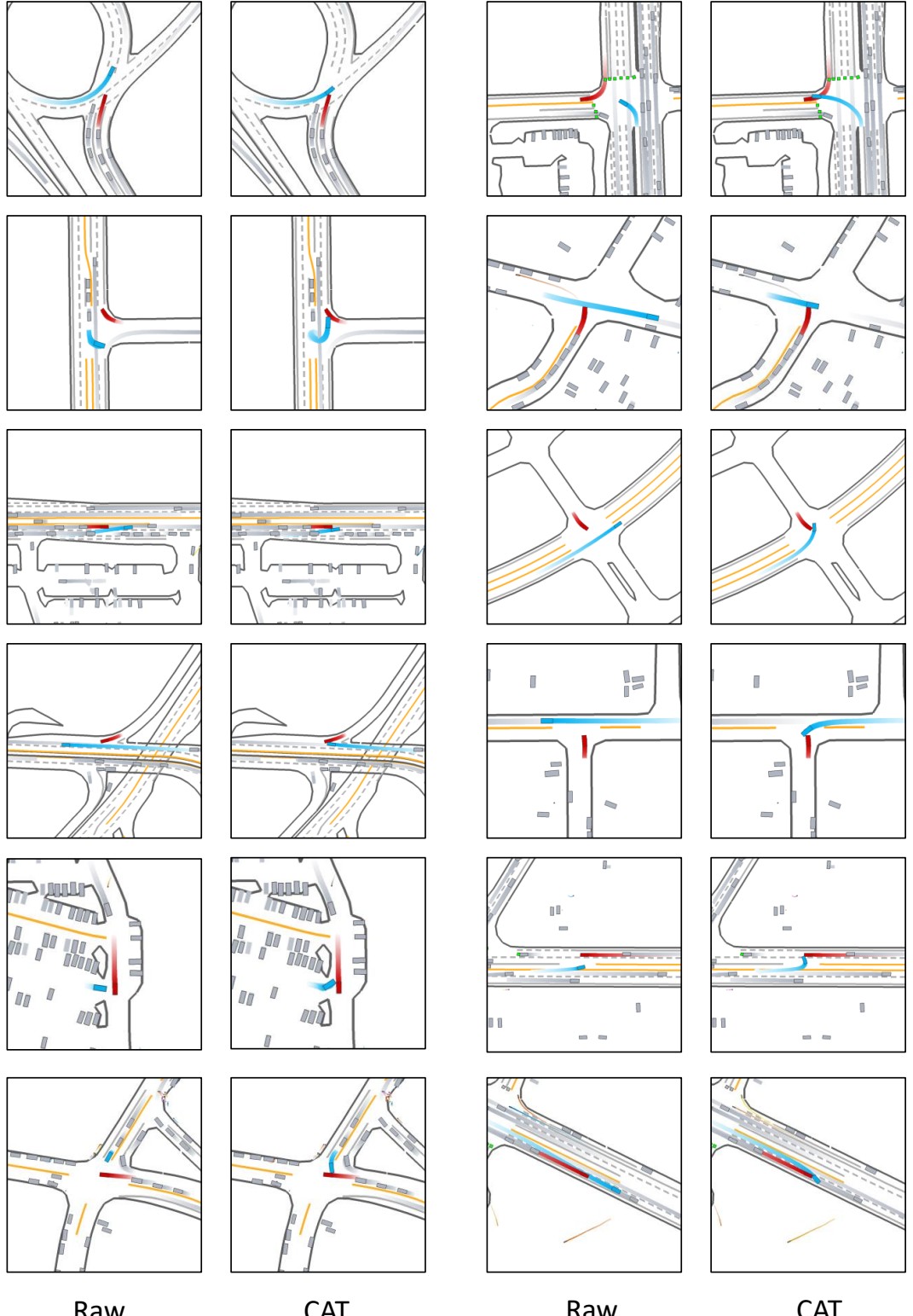

Raw          CAT          Raw          CAT

Figure 6: More comparison between the original scenarios in raw datasets and the safety-critical scenarios generated by CAT. The red car is the ego vehicle and the blue car is the opponent vehicle.

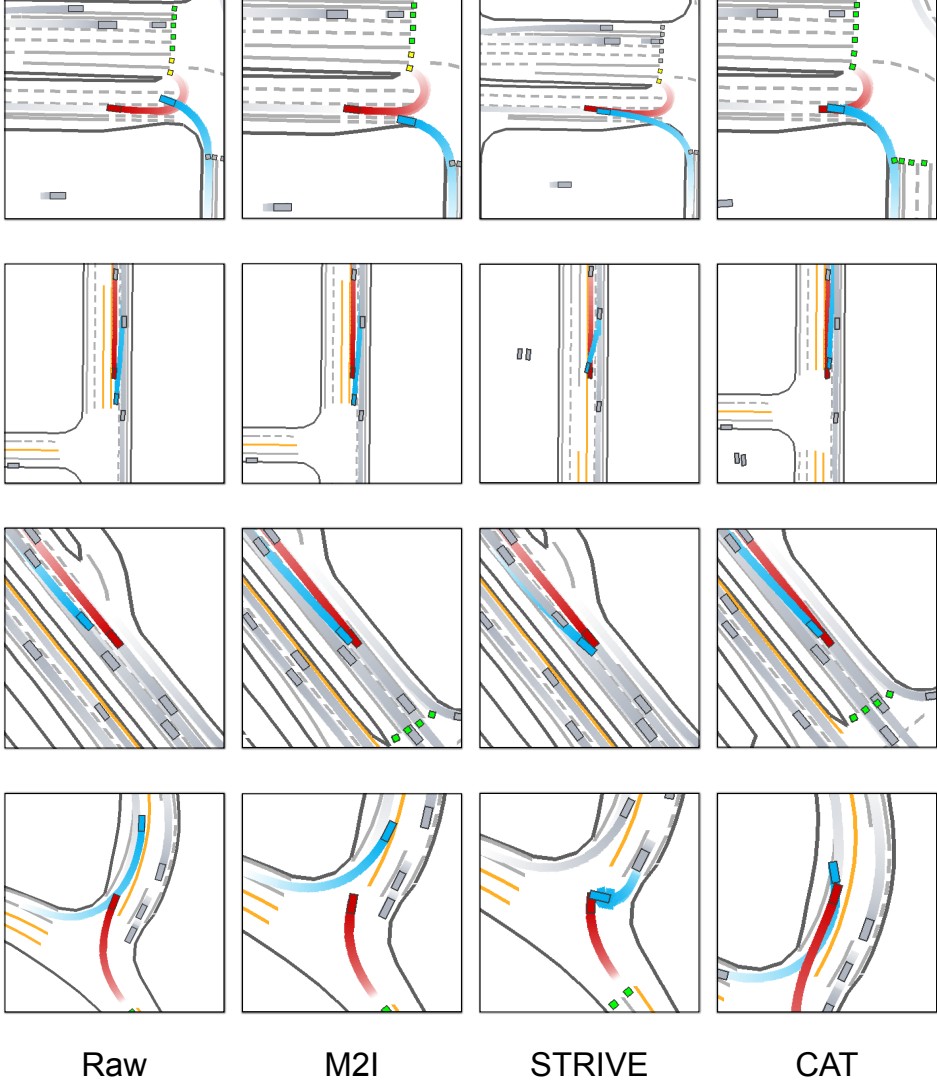

| Raw | M2I | STRIVE | CAT |

Figure 7: Comparing the different scenario generation methods. The red car is the ego vehicle and the blue car is the opponent vehicle.

## F  Details of the Rule-based Adversarial Traffic Generation

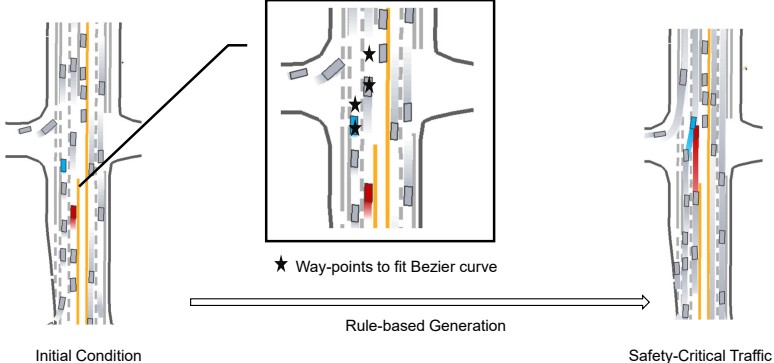

★ Way-points to fit Bezier curve

Rule-based Generation

Initial Condition                                                    Safety-Critical Traffic

Figure 8: An example of the rule-based adversarial traffic generation.

Considering the HD-map in Waymo datasets are highly unstructured, thus we design a rule-based system as follows:

1. We heuristically take the vehicle labeled as 'Object of Interest' as the adversary.
2. We take some waypoints on the navigation path of the ego vehicle, which will be occupied by the adversary later to minimize the ego vehicle's drivable area.
3. We mix above waypoints with those on the original path of the adversarial vehicle.
4. We fit a Bezier curve based on all the way-points to derive a smooth and feasible path of the rival vehicle.

## G   Qualitative Results of Safety Improvement after CAT

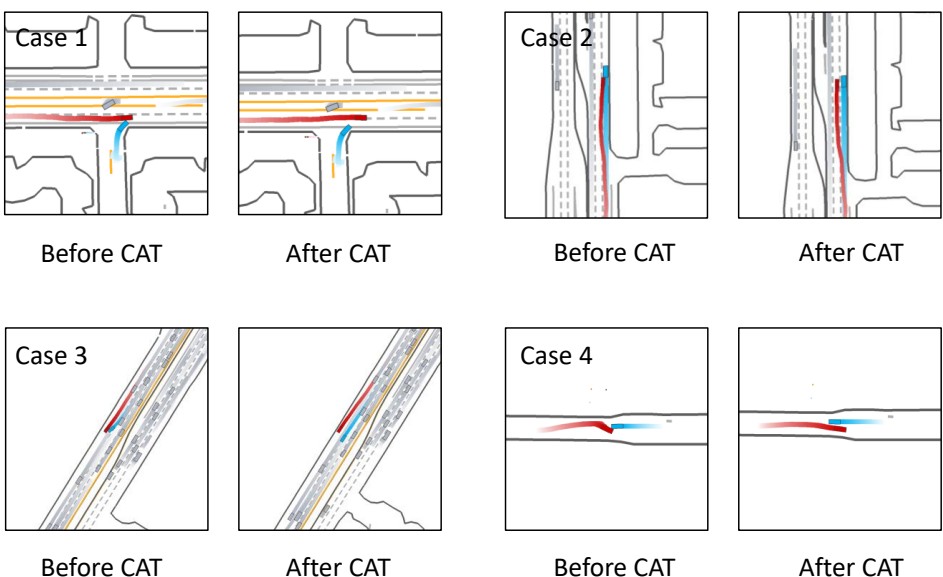

Figure 9: Driving behaviour before and after CAT. The red car is the ego vehicle and the blue car is the opponent vehicle. In case 1, the opponent car makes an unprotected left turn at an intersection; the driving agent learns to stay away from potentially dangerous vehicles. In case 2, the leading car slows down; the driving agent learns to change its lane and overtake. In case 3, the opponent car cuts into the lane suddenly; the driving agent learns to yield and change its lane ahead of time. In case 4, two vehicles traveling in opposite directions meet and the driving agents learns to pass by.

## H   Further Discussion

**Limitations:** Following limitations wait to be addressed in future work: (i) we only consider adversarial vehicles in this work but the safety-critical behaviors of pedestrians and cyclists are also of importance for safe driving and yet to be done, it requires the access to a different motion forecasting model; (ii) Experiment on five hundred scenes cannot cover all the accident-prone situations, thus there are other possible failure modes in the resulting agent; (iii) we only investigate the RL-based driving policy but the adversarial scenarios should also benefit the human-in-the-loop imitation learning [17, 43].

**Transferring to real-world driving:** The proposed adversarial training method and the comparison with prior methods are evaluated in the simulation of one hundred complex traffic scenarios imported from real-world driving dataset [4]. Thus, the evaluation contains realistic and complex vehicle interactions and shows promise for transferring to real-world settings.

