# OpenReview forum: "CAT: Closed-loop Adversarial Training for Safe End-to-End Driving"
_robot-learning.org/CoRL/2023/Conference — CoRL 2023 Poster_

### Official Review · Reviewer_s5WY · 2023-07-16

**Confidence:** 3
**Originality:** Good
**Technical Quality:** Very Good
**Clarity Of Presentation:** Very Good
**Impact:** 3

**Recommendation:**

Weak Accept: I recommend accepting the paper, but will not argue for my recommendation if the majority of other reviewers have a different opinion.

**Review:**

## General strengths and weaknesses
### Strengths

S1. Relevant topic (adversarial training for autonomous driving). The paper presents a useful approach to generate adversarial test scenarios from real data, as these scenarios are rare in real-world data sets.

S2. Video and plots help the reader to understand the framework and illustrate the results well

### Weaknesses
W1. The paper provides a framework for safe end-to-end driving, but does not explain or specify what safe end-to-end driving is

W2. Lack of a clear formulation of the control problem. In particular, the state and action spaces are unclear. As this is not a general reinforcement learning setting, but an explicit driving use case, these formulations are crucial to understanding the results (Factorized Safety-Critical Resampling)

W3. In the introduction, the paper claims that the setting is planner-agnostic. However, it just demonstrates how the framework can be used for RL. It is not specified how the framework would work for other policies. The Factorized Safety-Critical resampling builds on RL formulation and is not agnostic. Actually, the paper states this as a potential of future work (limitations section), so the setting is not agnostic and, hence, this is not a contribution of this work.


## Specific evaluation of requested categories
('S' - strength, 'W' - weakness)

### Clarity:
- S: The claimed contributions are stated clearly
- S: Overall, well organized
- W: The reinforcement learning problem formulation is unclear, especially the state formulation is unclear (see comments)
- W: It is not clear what is understood as safe end-to-end driving and what setting the paper uses

### Originality:
- The paper combines multiple existing components (MetaDrive, DenseNet, M2i probabilistic formulation) to create a more efficient framework for the generation of adversarial scenarios
- Related work is adequately cited
- It is clear how this contribution improves upon the related work

### Quality:
- W: Contribution 1) is not supported by any evidence
- W: Equation (2): it is questionable if it holds without further assumptions

### Significance:
- especially in the context of RL for autonomous driving, a more efficient framework to generate critical scenarios is relevant as these are rare in real-world data sets


## Comments:
The above evaluation is based on the following comments:

1) Abstract, line 13 "on hundreds of training scenarios" As far as I understood, the paper just uses 100 scenarios from the Waymo Open data set
2)  l. 22 Safe End-to-end driving is not further introduced
3) l. 50 How would imitation learning work in this setting, and how can IL react to adversarial attacks? In the limitations section the paper states:  human imitation learning in the loop. There is a difference between this and the general concept of imitation learning.
4) Comment on Problem Formulation/ Experimental results
   * It is not clear what the state and action spaces are.
   * In an end-to-end driving algorithm, the state should be raw sensor data (simulation from camera/lidar in Meta Drive (?) and the current position/ego motion data), and the action space should be steering angle and accelerator/ brake pedal position. In the paper, this is not clear. I assume the paper uses a similar setting as [19], Section 5.1. The state formulation should be included in the problem formulation.
5) Problem formulation: What is the destination?
6) Reward formulation: $R=d-\alpha c$ ,  $d$ should have a negative sign as a smaller distance to the destination should give more reward. Furthermore, the value range of $\alpha$ is not defined.
7) $\alpha$ is used twice (?) with different explanations: Problem formulation: "a hyper-parameter for the reward shaping" Implementation details: "a heuristic decay factor to reflect the uncertainty of traffic models with the increasing prediction horizon." It is unclear whether this is the same  or different parameters.
8) Equation (2) It is questionable if this holds, especially without knowing the state formulation. Given the setting, it is not obvious that this equivalence is always true
9) Evaluation of Closed-Loop Adveserial Training (CAT) (A) No Adv: remove the opponent vehicles: Where do the 7546 Train Attack Num come from if the opponent vehicle is removed?
10) limitations: the paper provides a framework to train driving agents but does not provide any safety guarantees, as it is agnostic to the planner. Work on the algorithmic side is needed to achieve safe driving and provide guarantees.
11) Publishing code would make this framework usable for others more easily

### Minor Comments / Typos
1) l.123 $\uparrow \downarrow$   introducing this notation would make it clearer
2) l. 125 $\sum R$ instead of  $\Sigma R$
3) l.138 $f^{Adv}$
4) l.119/ l. 149  $J(\pi, f^{Adv})$ instead of $J(\pi)$
5) Algorithm 1: BBox should be explained in a comment such that the pseudo-code can be understood by itself.
6) Table 2: "Replay" instead of "Heuristic" would be more consistent

**Quality Of The Limitations Section:**

Limitations are addressed clearly

**Questions For Rebuttal:**

### Suggestions for clarification during rebuttal
I suggest to reiterate the following points:
1. The paper should not claim the agnostic setting as a contribution
2. The paper should explain what is understood as safe end-to-end driving
3. The state and action spaces should be formulated clearly, and the used reward function should be explained more precisely ($\alpha$  and destination).
4. The paper should provide a more detailed proof for equation (2) or weaken the equivalence assumption.


### Questions for rebuttal
1. How would imitation learning work in this setting, and how can IL react to adversarial attacks?
2. Which setting of end-to-end driving is used? Which (sensor) data is used as an input?
3. Which state formulation was used? What is the action space?
4. What is the destination for the reward formulation? Can the reward function be explained more precisely? How can the parameter $\alpha$ be tuned?
5. Is it possible to prove that the two optimization problems in equation (2) are equivalent? Which assumptions are required for this?


===============================================================
### Response and update in response to rebuttal

I thank the authors for answering most of my questions and considering the feedback in the revised manuscript. I lifted my rating to "Weak Accept" as the criticism regarding the paper's clarity and technical quality were revised. I appreciate that the authors added the state formulation, clarified the safe end-to-end driving setting, removed the overstatement of the first contribution, and added further assumptions to Equation (2).

**Robotics Focus:**

Relevant but unlikely to deploy to hardware in near future

**Summary Of Paper:**

The paper provides a framework to train RL agents on accident-prone scenarios generated from real datasets and improve the agent's behavior in critical situations. Accident-prone scenarios are rare in real-world data sets. Therefore, CAT samples scenarios and adapts them to provoke accidents. Multiple trajectories of opponent vehicles are generated based on a real-world scenario. Based on these trajectories, the agent's reaction is simulated. In order to improve the agent's policy, the trajectory that is most likely to lead to a collision is chosen to create a scenario that is as critical as possible. The paper claims that the framework is agnostic to the learning method but only demonstrates this for one RL algorithm (TD3).
Experimental results comparing CAT to state-of-the-art methods show that an agent trained with adversarial scenarios performs better in real-world and accident-prone scenarios. Furthermore, scenario generation is less time-consuming than the SOTA method.

**Summary Of Recommendation:**

Overall this paper is a relevant contribution toward generating adversarial driving scenarios for autonomous driving. The structure is clear, and it is overall well-written.

However, there are some weaknesses regarding key aspects that are insufficiently addressed. My main points are:
1. I recommend clearly stating and specifying what the paper means by safe end-to-end driving and which setting is assumed.
2. The problem formulation is unclear, especially regarding the definition of the state and action spaces.
3. Even though the reinforcement learning algorithm is not a key contribution of this paper, the results are not reproducible without this information. Furthermore, the factorized safety-critical resampling technique, which is a contribution, builds on this formulation and cannot be validated without this formulation.

=========================================
Update after rebuttal

I changed my rating to "Weak Accept"  (see "Questions for rebuttal" for details).

---

### Official Review · Reviewer_P4zB · 2023-07-19

**Confidence:** 3
**Originality:** Good
**Technical Quality:** Very Good
**Clarity Of Presentation:** Very Good
**Impact:** 4

**Recommendation:**

Weak Accept: I recommend accepting the paper, but will not argue for my recommendation if the majority of other reviewers have a different opinion.

**Review:**

#### Quality
- The paper is technically sound. The claims are well supported by experimental results. The proposed method is practical and reasonably easy to implement.

#### Clarity
- The paper is clearly written and well organized.

#### Originality
- The problem is not new; the method is novel. The proposed method is clearly different from previous methods.

#### Significance
- The results are interesting, and support the main claims. Others are likely to use the ideas. Not significantly better than previous methods.

#### Strengths
- The paper is clearly written and easy to follow.
- The proposed method is practical and effective.
- Empirical results support the main claims


#### Weaknesses
- The ii) of the main contributions in the Intro says the method "balances attack success rate and computation cost by resampling ... ". However there is no experiment that demonstrate this point. An ablation study on how the method trades off success rate and compute cost would be helpful to support this claim.
- The dataset is very small (100 segments). (The author mentioned it in the limitation paragraph).
- In Fig 5. the learned policy is only evaluated in the MetaDrive simulator against log playback traffic with ~10% crash rate. This % is very high even given the small training dataset.
- Transferability to real-world autonomous driving systems: RL agent might overfit to the specific training environment, in this case the adversarial sim agents. The resultant policy could potentially be overly cautious and not make enough progress. It would be helpful to also plot the progress in Fig 5.


**Quality Of The Limitations Section:**

Additional details required

**Questions For Rebuttal:**

- The ii) of the main contributions in the Intro says the method "balances attack success rate and computation cost by resampling ... ". However there is no experiment that demonstrates this point. An ablation study on how the method trades off success rate and compute cost would be helpful to support this claim.
- The assumption of ego vehicle is unidirectionally conditioned on other vehicles might be too strong. Have the authors thought of relaxing this assumption (e.g. by rolling out the ego policy and other vehicle policies jointly in an autoregressive manner)? It might require reformulating the factorization.
- Transferability to real-world autonomous driving systems: RL agents might overfit to the specific training environment, in this case the adversarial sim agents. The resultant policy could potentially be overly cautious and not make enough progress. It would be helpful to also plot the progress in Fig 5.

Nice to have:
- The dataset is very small (100 segments). (The authors acknowledge it in the limitation paragraph). It would be nice to demonstrate similar results using the entire WOMD dataset.
- In Fig 5, the learned policy is only evaluated in the MetaDrive simulator against log playback traffic with ~10% crash rate. This % is very high even given the small training dataset. Curious if using a larger training set would further bring down the crash rate.


**Robotics Focus:**

Relevant but unlikely to deploy to hardware in near future

**Summary Of Paper:**

The paper proposed a novel iterative learning framework called Closed-loop Adversarial Training (CAT) to dynamically generate adversarial safety-critical scenarios for training a driving policy. The proposed method factorizes the safety-critical scenario generation objective into 3 components: 1) traffic prior 2) ego estimation and 3) collision likelihood. The method resamples the trajectories of the surrounding objects based on the whether they collide with the ego vehicle, while training an ego driving policy in an iterative manner. The paper provides both qualitative and quantitative experimental results to show superior performance of the proposed method compared to a few baseline methods on a driving set of 100 from WOMD.

**Summary Of Recommendation:**

The paper proposed a novel iterative learning framework called Closed-loop Adversarial Training (CAT) to dynamically generate adversarial safety-critical scenarios for training a driving policy. The method resamples the trajectories of the surrounding objects based on whether they collide with the ego vehicle, while training an ego driving policy in an iterative manner. The paper provides both qualitative and quantitative experimental results to show superior performance of the proposed method compared to a few baseline methods on a driving set of 100 from WOMD.

The paper is well written and easy to follow. The results are interesting and relevant to the field. Experimental setting is reasonable and clearly demonstrates the major claims of the paper. I recommend an accept condition on addressing the few minor concerns raised above.

---

### Official Review · Reviewer_9tii · 2023-07-19

**Confidence:** 3
**Originality:** Fair
**Technical Quality:** Fair
**Clarity Of Presentation:** Good
**Impact:** 3

**Recommendation:**

Weak Accept: I recommend accepting the paper, but will not argue for my recommendation if the majority of other reviewers have a different opinion.

**Review:**

- quality: overall paper is well written
- clarity: ideas expressed clearly
- originality and significance: I believe that even thought the idea might have originality, the experimental details can't fully confirm that it could actually works on practice for large scale training of autonomous driving, due to the following missing points:

is generated driving policy realistic? - factorized safety critical resampling is based on hypothesis that safety critical scenario can be identified as  a re-combination of possible, independent predictions (multiple-futures) where some of the re-combinations can be more safety critical than others,  - this doesn't consider scene-wise multimodal prediction, where most probable modalities have to match in realistic cases only (which is done via learning), thus the proposed generated safety-critical scenario can be mostly un-realistic therefore it will bias whole distribution of realistic scenarios in a closed-loop simulation framework

does experiment have fair split between train/val/test? - closed-loop adversarial training, doesn't have a mechanism to ensure that generated safety-critical scenarios are all meaningful (no distribution shift to unrealistic but safety-critical scenarios learning) and the experimental setup doesn't specify enough details to ensure that CAT produces generalized driving policy , which needs to be validated by ensuring that there is no overlap between training and test data (different locations/maps/traffic conditions/time, etc...)

**Quality Of The Limitations Section:**

Limitations are not well addressed

**Questions For Rebuttal:**

- factorized safety critical resampling is based on hypothesis that safety critical scenario can be identified as  a re-combination of possible, independent predictions (multiple-futures) where some of the re-combinations can be more safety critical than others,  - this doesn't consider scene-wise multimodal prediction, where most probable modalities have to match in realistic cases only (which is done via learning), thus the proposed generated safety-critical scenario can be mostly un-realistic therefore it will bias whole distribution of realistic scenarios in a closed-loop simulation framework

- closed-loop adversarial training, doesn't have a mechanism to ensure that generated safety-critical scenarios are all meaningful (no distribution shift to unrealistic but safety-critical scenarios learning) and the experimental setup doesn't specify enough details to ensure that CAT produces generalized driving policy , which needs to be validated by ensuring that there is no overlap between training and test data (different locations/maps/traffic conditions/time, etc...)

- not clear, which is of the proposed contributions has biggest impact on the ability of the CAT to generate safe driving policy, need to decompose the key contribution factors and how they improve the metrics/scores

- need to introduce baseline rule-based system to generate safety-critical scenarios, especially since HD-map information is used as a traffic prior, it should generate complete set of safety-critical scenarios given human replay as an input then compare the experiments with the baseline

**Robotics Focus:**

Relevant but unlikely to deploy to hardware in near future

**Summary Of Paper:**

paper presents "CAT: Closed-loop Adversarial Training for
Safe End-to-End Driving", where the goal is to augment training set of replay scenarios with automatically generated safety-critical scenarios in closed-loop simulation to train safe driving policy

**Summary Of Recommendation:**

I recommend to re-enforce the experimental section with fair train/test/val split and introduction of the rule-based baseline with traffic-priors to confirm the claim of the paper at the rebuttal stage, so that the contribution of the authors could be fairly judged

---

### Official Review · Reviewer_hXmh · 2023-07-22

**Confidence:** 4
**Originality:** Very Good
**Technical Quality:** Very Good
**Clarity Of Presentation:** Very Good
**Impact:** 4

**Recommendation:**

Strong Accept: I recommend accepting the paper and will argue for my recommendation even if other reviewers hold a different opinion.

**Review:**

= Contribution

 The paper tackles a research topic which is linked to distributional shift in E2E training and deployment. This is maybe the main challenge in E2E development now. How do we tackle distribution shift to long tail event that typically are rare during training. This contribution is quite straight forward in the way that it provides a simple decomposition mechanism for generate long tail scenarios. The fact that it can generate these scenarios at a rate that is adequate for training makes it quite appealing.

= Research Idea

== Quality

The paper is very clear and well written. The idea is well explained and well argued. Limitation of the paper are quite well justified. The paper could have had more in depth analysis with more scenarios and checking of the policy robustness after training.

== clarity

The paper is clear.

== originality

The paper is rather original. The contribution is well put together.

== significance

This work is significant and proposes a solid starting point for performing adversarial scenario sampling during trail and error learning in simulation.
More could be done in terms of convergence to an optimal policy. The number of crashes in these context continues to grow through training. One would except that with more rollout the agent learns to avoid all corner cases.

Strengths
-	Good results
-	Some baseline comparison

Weakness
-	Could have more indepth statistical analysis
-	More scenarios


**Quality Of The Limitations Section:**

Limitations are addressed clearly

**Questions For Rebuttal:**

- Could we get the some robustness evaluation of the best policy?

**Robotics Focus:**

Highly relevant to robotics but no hardware experiments

**Summary Of Paper:**

This paper presents a reinforcement learning agent, that can generate adversarial scenarios during training. This is done by decomposition the scenario generation process in three factors. First the traffic prior which is simply based on a generic motion prediction model. Second the ego vehicle density, which is based on the currently learned policy. And third some safety critical index, for instance linked to geometrical proximity of the agents in the scene. The proposed framework allows to generate scenarios for simulation-based learning that are adequate with respect to computational requirement of reinforcement learning. The authors show in their experiments that the resulting policies are more robust than regular policies.

**Summary Of Recommendation:**

- Put more emphasis on robustness at convergence

---

### Author Response · Authors · 2023-08-10
**Highlights of the revised manuscript**

We thank all the reviewers for their time and valuable suggestions. We have uploaded the revised manuscript based on the reviewers’ comments. We highlight our major updates as follows:

- We add an anonymous code link (see Abstract, line 18) and will make the code publicly available for reproducibility.
- We rephrase our key contributions to avoid the misleading and over-claiming (see Section 1, line 65-70)
- We clarify our safe end-to-end driving problem formulation (see Section 3.1, line 109-118).
- We clarify our experimental settings of RL (see Section 4, line 202-204; Appendix C).
- We enforce the experiments with
    - progress plots corresponding to crash rate plots (see Fig. 5)
    - robustness analysis under physical attacks during the evaluation (see Tab. 2)
    - a rule-based safety-critical scenario generation baseline comparison (see Tab.2 and Appendix F)

We welcome the reviewers to check them in the revised manuscripts to ensure that their concerns are properly addressed.

---

### Author Response · Authors · 2023-08-15
**Any feedback on the rebuttal?**

Dear Reviewers and AC,

Can you please take a look at the rebuttal we have spent a lot of time preparing and give us some feedback? Thank you.

Authors

---

### Decision · Program_Chairs · 2023-08-30

**Decision:**

Accept (Poster)

**Comment:**

This paper presents an interest adversarial training scheme for autonomous vehicles, which takes in real-world safe driving trajectories (states, no sensor observations) and modifies them locally to create adversarial behaviors, based on a collision-maximizing objective. The resulting optimized adversarial behaviors are set as the open loop behavior of opposing (non-ego) agents, aiming to stress-test the current ego-vehicle's policy that is being trained. The paper shows that this adversarial training scheme results in slightly more robust ego-policies than training based on logged safe trajectories for opposing agents. The paper also shows that this scheme is computationally much more efficient than other similar techniques in this area, such as STRIVE [13].

Overall, reviewers found the idea interesting, but were concerned about the lack of realism of the attacks, no hardware experiments, as well as the fact that although the attacks are computationally efficient, they are not much more effective at producing adversarial traffic scenarios than [13]. If [13] had used a generative model with cheaper inference than an autoregressive model, reviewers were concerned that the impact of the CAT paper would be relatively small.

In any case, I think the idea is interesting enough to discuss at the conference. I am recommending acceptance as a poster.

[13]  D. Rempe, J. Philion, L. J. Guibas, S. Fidler, and O. Litany. Generating useful accident-prone driving scenarios via a learned traffic prior. In Proceedings of the IEEE/CVF Conference on Computer Vision and Pattern Recognition, pages 17305–17315, 2022.